# Recent Advances of Doped SnO_2_ as Electron Transport Layer for High-Performance Perovskite Solar Cells

**DOI:** 10.3390/ma16186170

**Published:** 2023-09-12

**Authors:** Vo Pham Hoang Huy, Thi My Huyen Nguyen, Chung Wung Bark

**Affiliations:** Department of Electrical Engineering, Gachon University, Seongnam 13120, Gyeonggi, Republic of Korea; vophamhoanghuy@yahoo.com.vn (V.P.H.H.); myhuyen.dho.k12@gmail.com (T.M.H.N.)

**Keywords:** perovskite solar cells, tin oxide, electron transport layers, doping materials

## Abstract

Perovskite solar cells (PSCs) have garnered considerable attention over the past decade owing to their low cost and proven high power conversion efficiency of over 25%. In the planar heterojunction PSC structure, tin oxide was utilized as a substitute material for the TiO_2_ electron transport layer (ETL) owing to its similar physical properties and high mobility, which is suitable for electron mining. Nevertheless, the defects and morphology significantly changed the performance of SnO_2_ according to the different deposition techniques, resulting in the poor performance of PSCs. In this review, we provide a comprehensive insight into the factors that specifically influence the ETL in PSC. The properties of the SnO_2_ materials are briefly introduced. In particular, the general operating principles, as well as the suitability level of doping in SnO_2_, are elucidated along with the details of the obtained results. Subsequently, the potential for doping is evaluated from the obtained results to achieve better results in PSCs. This review aims to provide a systematic and comprehensive understanding of the effects of different types of doping on the performance of ETL SnO_2_ and potentially instigate further development of PSCs with an extension to SnO_2_-based PSCs.

## 1. Introduction

Perovskite was identified in 1839 with the regular formula ABX_3_, where the unique structure of close-packed oxides can contain a very large cation. Thus, compared to inorganic cations, the larger ionic radii of small organic cations cause the creation of an organic–inorganic hybridization state in the perovskite structure [1,2,3,4,5]. Perovskite solar cell (PSC) technology has captivated worldwide attention because of the remarkable advances in power conversion efficiency (PCE), which has increased rapidly from 3.7% to a verified 25.2% within just a decade [2,6,7,8,9,10,11,12]. Among the components of PSC, the ETL plays a significant role in the transportation and extraction of photogenerated electrons, regulation and modification of energy levels, and the reduction in electrical recombination accumulates in the network [13]. The evaluation of an efficient ETL requires high electron affinity, ion potential and high electron injection efficiency, while simultaneously suppressing the recombining holes at the electrode interface of ITO or FTO.

Currently, n-type metal oxide ETLs are commonly used in PSCs with certified proven performance. The electronic characteristics of various metal oxide ETLs (namely energy level (eV) and bulk mobility (cm^2^ V^−1^ s^−1^)) are summarized in Figure 1. The ETL materials with high electron mobility are clearly favorable due to reduced probability of charge recombination as well as minimized charge accumulation at the surface interface. The structure of a common PSC is illustrated in Figure 2a. Among these, mesoporous TiO_2_ is the most commonly employed ETL material because of its high transmittance and suitable band gap, which ensures the high performance of PSC. However, the high annealing temperature required to obtain high-quality compact TiO_2_ films hinders their applicability in flexible devices, incurring increased production costs. In addition, the difference in the mobility of the perovskite (7.4 cm^2^/V·s) and TiO_2_ (0.15–4.1 cm^2^/V·s) materials diminishes the charge transportation capacity of the device [14]. ZnO is also used as an ETL in PSC; however, low-temperature-processed ZnO undergoes OH^−^ residual on the external surface, which disintegrates perovskite absorbers [15,16,17]. In addition, metal oxide ETLs such as Cr_2_O_3_, WO_3_, CeO_2_, In_2_O_3_, and Nb_2_O_5_ have not been extensively studied as there is no clear disparity in advantage when compared with the most commonly used device TiO_2_ [18,19,20,21,22]. 

Ever since PSCs were first reported, an SnO_2_-based ETL has been mentioned as the best electron transport layer in devices. Recently, extensive research has been conducted to control the morphology of SnO_2_ to maximize the light-absorbing interface area of perovskite–SnO_2_. The number of studies on the ETLs commonly used in PSCs (including TiO_2_ and SnO_2_) is presented in Figure 2b. Although there have been few studies on SnO_2_ ETLs at baseline, there has been a continuous increase in studies. This is explained by the advantages of SnO_2_ when studied as an ETL in PSC (Figure 2c). In addition, owing to its excellent chemical stability and simple synthesis, SnO_2_ is used as a typical sensing material, particularly in formaldehyde sensors [23,24]. With the recent changes in the composition of perovskites, studies have focused on surface engineering to tune the energy level and control the electron pathway characteristics of SnO_2_. Doping-based SnO_2_ is an effective solution for enhancing the electron transport capacity and interface properties of ETL, leading to improved electron mining in PSCs.

Herein, the influence of impurities on the electronic properties and how they affect the morphology of SnO_2_, that is, the intentional incorporation of impurities into the SnO_2_ network, are discussed. Doping is a well-known method for increasing the charge separation, thereby promoting the spectra in the solid-state solar cell [25,26,27]. Based on the evaluations, the properties for selecting the right doping type, as well as the location where they affect the crystal lattice of SnO_2_, are crucial factors in improving the performance of PSC. Another way to control recombination is by binding to organic compounds, such as polymers, which is also within the scope of this review. To begin with, the primary factors affecting the ETL properties are listed. This is followed by an outline of the special properties of SnO_2_ used as an ETL. Subsequently, the detailed mechanisms for doping selection based on their suitability for different synthetic techniques are explored. Then, we summarize the takeaways from these doping studies and the ways in which they affect SnO_2_ properties to achieve better efficiency in PSCs. Finally, the enormous potential to use doped SnO_2_ as an ETL in PSCs is evaluated.

## 2. Inorganic–Organic Halide Perovskite Solar Cell

### 2.1. Device Configuration and Working Mechanism of Halide Perovskite Solar Cells

#### 2.1.1. Device Configuration

The solid-state PSCs have five essential components: the transparent conductive oxide (TCO) such as fluorine-doped tin oxide (FTO) or indium tin oxide (ITO), electron transport material (ETM), the light-harvesting perovskite material, hole transport material (HTM), and back metal contact (e.g., Au and Ag). Depending on the difference in morphology, PSCs are classified as two main types of devices including mesoscopic-type PSCs and planar-type PSCs (as illustrated in Figure 3). The mesoscopic PSCs usually employ mesoscopic oxides (TiO_2_, ZnO, or even insulating oxide of Al_2_O_3_) that act as a scaffold layer [28,29]. Thereby, perovskite solution can both infiltrate the pores and cover the surface of the mesoporous scaffold, consequently forming a homogeneous film. Moreover, generated electrons from perovskite crystals can extract and transfer quickly to the electrode. However, the preparation of mesoscopic oxide requires a high temperature (about 450–500 °C) for sintering, leading to the consumption of energy as well as causing several detrimental effects on the device in heat treatment. Therefore, researchers removed the scaffold to make a flat perovskite layer between the electron transport layer (ETL) and hole transport layer (HTL), which is named planar-type PSCs. The interest in the planar PSCs is that perovskite crystals exhibited larger grain sizes and fewer pinholes than mesoscopic ones because the perovskite grain growth is unrestricted by the pore size of the scaffold as in the mesoscopic assembly [30,31]. In addition, the thickness of layers and the interfaces between them can be simply controlled. Basically, both mesoscopic-type PSCs and planar-type PSCs become instantly drawn to researchers, and both demonstrated high PCEs (over 22%). Additionally, in two types of PSCs, the active layer of perovskite is sandwiched between the electron transport layer (ETL) and the hole transport layer (HTL). Depending on the structure or the growth sequence of constituent layers, where the ETL or HLT is deposited onto the TCO first, the PSCs are further categorized as conventional (n-i-p) and inverted (p-i-n) devices, respectively. For the two mentioned device configurations, the selection of charge transport materials (ETM and HTM) and charge collection (anode and cathode) are crucial in order to be consistent with the band energy.

#### 2.1.2. Working Mechanism of Halide Perovskite Solar Cells

Regardless of the architecture, the working mechanism in PSCs can be generalized as depicted in Figure 4. Briefly, (1) generation of charge carriers: When PSCs are irradiated under solar light, the perovskite layer absorbs photons to produce e h pairs (excitons). The next step is the separation of charge carriers to form free electrons and holes due to the difference in the exciton binding energy of perovskite material. After that, free charge carriers are transferred in two directions including the (2) electron transfer process: free electrons in the excited state of the conduction band (CB) of perovskite layer are injected into ETM, meanwhile, the (3) hole transfer process: the positive holes in valance band (VB) of perovskite layer are injected into HTM. Finally, charges are collected at the corresponding electrodes, then the current is generated in the outer circuit when TCO and metal electrodes are connected. While charge separation generates the current, electrons and holes can recombine into excitons such as radiative carrier recombination (4) or non-radiative carrier recombination (5), which lead to energy losses. Additionally, a series of charge recombination at (6) ETL/perovskite and (7) HTL/perovskite interfaces due to the reverse transmission of electrons/holes before migrating to the electrodes and the (8) ETL/HTL interface due to the insufficient coverage of perovskite layer are detrimental to the performance of PSCs [32,33]. Therefore, with the aim of reducing charge recombination, as well as achieving high performance, it is noteworthy to study the morphology and property of constituent layers, simultaneously focusing on enhancing interfacial contacts of interfaces.

### 2.2. Challenge in the ETL of PSC

Characterization of the ETL in the PSC device plays a significant role in achieving a high PCE. In addition to affecting the collection and transportation of charge, the ETL also acts as a hole-blocking coating at the interface to limit electron–hole recombination. Therefore, in addition to regulating the open-circuit voltage, the ETL also affects the fill factor of a solar cell. Metal oxides have emerged as prospective replacements for the conventional structure of suspended TiO_2_ due to their alluring photochemical properties, with SnO_2_ emerging as the most suited choice due to the following exceptional benefits: (i) to simplify charge transport and improve ohmic contact, SnO_2_ has a deeper conduction band than TiO_2_ and a wider band (3.6–4.1 eV) [34,35,36]; (ii) SnO_2_ has better band alignment with perovskite absorbers and has higher electron mobility than TiO_2_ [37,38,39,40]; and (iii) SnO_2_ ETL can be deposited in solution treatments with low temperature and no high sintering temperature requirement of the compact TiO_2_ ETL [41,42]. In principle, the significant agents that ascertain the final commercialization of ETL require the following criteria to be satisfied: (i) high mobility of electronics, contributing to meeting the mining demand, efficient charge extraction from the perovskite layer, and reduced charge recombination; (ii) efficient charge transfer and hole blocking require appropriate energy level alignment, avoiding loss of resistance and improving electron transport from perovskite to the ETL; (iii) opto-energy loss is decreased by high optical transmittance; (iv) ultimately satisfy high stability, low cost, and easy processing [1,43,44]. Nevertheless, for PSCs to achieve outstanding performance, ETLs must satisfy the conditions of the temperature process, ETL–perovskite interface, hysteresis behavior, and ETL thickness (as shown in Figure 5).

#### 2.2.1. Temperature Process

ETLs in organic–inorganic hybrid PSCs require thermal sintering to achieve good quality and crystalline state in a solution of perovskite precursors. The fabrication of long-life PSC requires excellent stability of both the charge-transfer material and perovskite-contact interface. Moreover, the lack of thermal stability of PSC impedes its capacity to be widely utilized in consumer electronics. Therefore, it is crucial to determine the appropriate temperature range for increasing the charge of the selective and carrier layers in PSCs [45,46,47,48]. For ETLs with low and unstable mobility and conductivity defects under light (in particular TiO_2_), high-temperature (>500 °C) preparation is required. ETL synthesis at high temperatures limits the commercialization and development of portable PSCs. In comparison, the excellent properties of SnO_2_, such as its chemical stability and excellent electronic mobility, enable SnO_2_ to be used as ETLs with relatively low-temperature synthesis [49]. This simple low-temperature process is suitable for the low-cost production of PSCs on flexible substrates. Furthermore, the low-temperature synthesis of metal oxides is also advantageous for obtaining high optical transparency, energy levels adjacent to the maximum valence band (EVB) or minimized conduction band (ECB), and high electronic transport costs, all of which contribute to the stability of PSC devices. Therefore, low-temperature synthesis of SnO_2_ is favorable for applications in high-performance flexible PSCs. For the low-temperature synthesis of SnO_2_, methods that entail the thermal disintegration of Sn-based precursors and direct spin coating of the SnO_2_ solution are most commonly used. Del Gobbo et al. synthesized SnO_2_ based on facile chemical bath deposition (CBD) at a very low temperature (<60 °C). The synthesis method proved to be effective without post-annealing treatment, which is needed to remove Cl^−^ residue from tin (IV) chloride precursor, resulting in PSC that yielded the required PCE of 14.8% [50]. As a result, a 20 nm uniform surface of SnO_2_ was formed without pinholes (as shown in Figure 6a,b). Figure 6c shows the moderate roughness of the film surface with Rq = 5.4 nm, compared with the roughness of the compact TiO_2_ surface at Rq = 16.1 nm. Furthermore, the low-temperature process of SnO_2_ showed an amorphous state, and there was no diffraction-associated tetragonal rutile SnO_2_ (Figure 6d). Moreover, from Figure 6e, it can be seen that a large barrier is formed to effectively impede imaging holes through the valence band of 2.7 eV at the ETL/perovskite interface. The author assumes that the major barrier that appears here is strongly dependent on the degree of conversion and the actual processing conditions. The change in the crystal structure of SnO_2_ enhanced the electron transport pathway of the perovskite film. In addition, Liu et al. minimized surface defects and improved the crystal quality of SnO_2_ ETL via low-temperature (<90 °C) solvothermal synthesis, which promotes process simplification and cost-effectiveness. The device possessed a PCE up to 19.21% and excellent stability after 1440 h in atmospheric environments (more than 93% retention) [51]. Nevertheless, stoichiometric SnO_2_ possesses morphological insulating properties and degenerates n-type semiconductor properties; therefore, the SnO_2_ conductivity in the primary layer is comparatively low when synthesized at low temperatures, leading to the system limitations, current density (Jsc), and fill factor (FF) of PSCs. Furthermore, SnO_2_ formed at low temperatures is an insulating material with poor wetting permeability relative to the perovskite precursor solution. Therefore, it is necessary to find a solution to support the synthesis of materials at low temperatures to further improve PCE performance. Recently, doping materials have been selected as potential solutions to further enhance the electronic conductivity of SnO_2_ films, even at low temperatures. Details of the use of doping materials are presented in the following sections.

#### 2.2.2. ETL/Perovskite Interface

PSCs are often formed in a normal or inverted conformation, where the normal structure is established when the FTO or ITO substrate is deposited on an ETL; in contrast, the substrate is bound to the HTL, and an inverted structure is formed (as shown in Figure 7a). The common interfaces exist in each structure: FTO (or ITO)/ETL/perovskite/HTL/Au (or Ag) interfaces for the normal structure and Au (or Ag)/HTL/perovskite/ETL/FTO (or ITO) interfaces for the inverted structure. However, interface engineering with HTL is beyond the scope of this review. For the interface between FTO (or ITO) and the ETL, the conductivity and transparency of the interlayer need to be considered. Transparent properties and high electrical conductivity are vital for this purpose. Simultaneously, the requirements for higher physicochemical properties by both FTO (or ITO) and ETL should also be considered, whereas the ETL/perovskite interface requires good charge separation; in other words, modifying engineering at the interface of ETL/perovskite is required to improve charge carriers and charge separation [52]. The defects, energy barriers at the perovskite/ETL/FTO (or ITO) interface, electrons in the perovskite, charge transport layers, and charge mobility affect the charge efficiency and latency in PSC devices. The defects are formed because of the difference in the lattice constant between the substrate and the coating in the PSC, while the energy barriers present at the interface easily cause charge accumulation and recombination. Moreover, the reaction that occurs between water and the molecules in the air and active layers is caused by defects, which eventually cause severe degradation of the perovskite layer. Therefore, interfaces should be handled carefully to avoid reducing the efficacy of the PCE. 

An efficient interface requires selective rejection of minority carriers and extraction of majority carriers. Specific to the ETL/perovskite interface, the band difference at the interface of the ETL and perovskite layer results in a discontinuity of the conduction band, which is conducive to the formation of an energy spike or cliff (ΔE_c_) at the interface. As depicted in Figure 7b,c, for a cliff structure, defects exist that act as recombination centers for charge mobility at the interface, in which electrons and holes cancel each other in a forward-biased state, which results in a barrier crossing and returns to the carrier interface. By contrast, the significant increase in ETL electrons in the barrier (ΔE_c_) results from the spike at the interface. Consequently, electrons get injected back into the interface in the polar direction [53]. At the FTO (or ITO)/ETL interface, the substrate is typically washed with a series of solvents (acetone, isopropanol, deionized water, and ethanol) to clean the surface prior to coating the ETL material. However, at the ETL/perovskite interface, the ETL surface often undergoes pretreatment (e.g., plasma treatment) with perovskite precursor solution to increase the wetting properties, resulting in increased cost and unsuitability for scaling production. From Figure 7d,e, it is apparent that the mechanism to achieve the desired performance is to correct the energy difference between the perovskite and ETL, specifically V_oc_. V_oc_, depends on the difference between representing electrons (EFN) and representing holes (EFP), where EFN tends to decrease with the minimum conduction band (CBM), while EFP moves to a site adjacent to the valence band (VBM) when illuminated. Therefore, it is crucial to construct the correct cascade of layers to create an efficient interface, resulting in carrier exploitation without loss of power. The large energy offset was caused by electron back-transfer recombination at the interface. In contrast, a small energy offset causes charge accumulation at the interface because of the formation of a weak electric field and incomplete migration of charge carriers. Therefore, optimizing interface engineering remains a major challenge for achieving high-performance PSCs.

#### 2.2.3. Hysteresis Behavior

In recent years, PSCs have been known for their outstanding performance; however, PSCs still endure severe hysteresis and unstable output under typical operating conditions [54,55]. The J–V hysteresis behavior is concerning for the development of PCE. The main proposed mechanisms for the J–V hysteresis of PSCs comprise the dynamics of charge carrier trapping and de-trapping processes, band bending, and slow transient capacitive current owing to ion migration or polarization of the ferroelectric. Ion movement is considered to be the primary cause of the J–V hysteresis and is indicative of cell degradation [56,57,58]. In theory, perovskite films allocate charge carriers and voids in various pathways to the ETL/perovskite interface owing to forward/reverse bias settings. In addition, electrons are directed to the n-type doped surface of the membrane to preserve charge neutrality in the layer owing to the accumulation of positive ions at the surface. By contrast, the accumulation of gaps equivalent to negative charges on the opposite side leads to p-type doping. E_a_ can be considered to distinguish which type of ion mainly causes J–V hysteresis in PSCs. Theoretically, when comparing the E_a_ of ion migration or vacancies with that of hysteresis, these two factors constitute the main components causing hysteresis. In summary, hysteresis behavior is formed by different effects on ion migration under different conditions. To be more specific, the charge carriers can migrate in a certain direction as a result of the built-in electric field, which creates a local electric field that can partially cancel the built-in electric field. Under reverse bias, electrons and vacancies move even deeper, enhancing the shielding of the built-in electric field and impeding the extraction of charge carriers. Under forward bias, the local electric field declines or even vanishes, favoring carrier extraction. The J–V hysteresis behavior can be limited via improved toll collection at the interfaces in PSCs, regardless of the governing mechanism. The origin of the J–V hysteresis was further investigated by Yan et al. [59], and it was reported that J–V hysteresis behavior occurs owing to an imbalance in the charge carriers of the ETL (or HTL)/perovskite interfaces. In addition, poor pretreatment of the ETL causes unbalanced charge transport, leading to the poor electrical conductivity of PSCs. Therefore, to improve the J–V hysteresis, the current through the ETL (or HTL)/perovskite interface should be stable. This requires good contact between the ETL and the perovskite. A significant potential drop occurs at the backside perovskite/HTL junction when ETL/perovskite performance is poor. Additionally, annealing the ETL under atmospheric conditions significantly reduced the J–V hysteresis. Consequently, high-efficiency plane PSCs with a stable output power of up to 20.4% are reached. Research has shown that current stability at the interface is a major obstacle to achieving high-performance PSCs. Based on this inspiration, Zhao et al. proposed the use of a PEG/PCBM organic scaffold to enhance PCE performance (Figure 8a) [55]. SEM analysis was conducted (Figure 8b–h) to understand the roles of PCBM and PEG. The addition of PCBM has little effect on the structure of the perovskite film; however, the film has large pinholes that pose a short-circuit risk in the absence of PEG. By contrast, the films with PEG exhibited a homogeneous substrate. Thus, PCBM supports passive charge trapping on grain boundaries and forms an electron transport pathway to the ETL, contributing to charge transport stabilization while PEG improves perovskite exposure through the formation of a compact and dense matrix on the layer, resulting in a PCE of 17.1%.

#### 2.2.4. ETL Thickness

Charge carrier transport depends on the morphology of the different interfaces in the PSC, thus modifying the ETL/perovskite interface morphology to facilitate electron transport, thereby achieving the desired performance. With an ETL thickness > 100 nm, the PSC can be affected by hindering the electron transport ability of electrons having to travel a longer distance to reach the top electrode and by increasing the recombination rate, which affects the cell filling factor and efficiency [60]. In contrast to the ETL thickness of <30 nm, the film experienced better light transmission efficiency owing to its high transmittance, small leakage current at the RTL/perovskite interface, and low recombination rate [61]. However, thin films present a technical problem owing to their severe unevenness, which contributes to a gradual increase in unwanted interface errors [62]. Therefore, a film thickness of 40 nm proved to be ideal for a high fill factor.

Furthermore, as described in [63], each precursor may need a different thickness to achieve the best cell function because the material properties differ intrinsically from the morphological features. For instance, titanium isopropoxide (c-TTIP), titanium diisopropoxide Bis(acetylacetonate)(c-TTDB), and tetrabutyl titanate (c-TBOT) were compared by Qin et al. [64], and c-TTIP was shown to be the best. In Figure 9a–d, the c-TTIP film has a moderately smoother surface than that of c-TBOT and c-TTDB. The roughness of the bare FTO is 13.5 nm, and following coating with c-TTIP, c-TBOT, and c-TTDB, it became 6.65, 9.38, and 11.4 nm, respectively. The substrates were significantly more consistent and smoother following the treatment with c-TiO_2_. This implied that the FTO/TiO_2_ layers were successfully coated. However, the thickness of the layers varied significantly. Additionally, mesoporous TiO_2_ was present during the confrontation of ETLs, which may have marginally disrupted the effects of thickness, and the surface roughness of the FTO employed in these studies was approximately two times lower than that of the typically used materials. Choi et al. examined the connection between the morphology of a typical ETL on FTO and the characteristics of electron mobility [65]. A reduction in the area of the interface at the perovskite/ETL/FTO and poor electron mobility were caused by the spin-coating method, which created a thick, highly irregular S-TiO_2_ film on top of the FTO rough layer. Therefore, increasing the thickness caused an increase in the ring surface of the ETL film (Figure 9e–j). The film thickness was also evaluated. Implementing an S-TiO_2_ ETL using a straightforward anodization and sputtering process solves these issues. The resulting S-TiO_2_ ETL exhibited remarkable physical properties, including the absence of pinholes, single-crystalline properties, uniform film thickness, increased transmittance, and strong linkages at the FTO/ETL interface. It also exhibited a well-defined nanostructured morphology. The length of the electron transport channel might further increase the thickness of the S-TiO_2_ ETL. The trapped electrons merged with the holes after being trapped in the thick S-TiO_2_ ETL (Figure 9k).

## 3. Crystal Structure, Electronic Properties, and Optical Properties of SnO_2_ and Doped SnO_2_

### 3.1. Intrinsic Semiconductor SnO_2_

The effective optical and electrical properties, stability band alignment, and excellent optical transparency make SnO_2_ a desirable candidate for commercial applications in solar converters and photovoltaic cells [66,67,68].

Due to the absence of free carriers, the intrinsic semiconductor SnO_2_ is not capable of conducting electricity; however, the use of SnO_2_ as an ETL in PSC is based on the outstanding features of deep conduction and valence bands, high transparency, wide bandgap, chemical stability, excellent mobility, high optical properties, and straightforward preparation at low temperatures [69]. The tetragonal rutile structure of natural cassiterite SnO_2_ is currently used in PSCs (Figure 10a). Rutile SnO_2_ belongs to the tetragonal system and possesses tetragonal space group D_4_h_14_ symmetry with lattice parameters γ = β = α = 90°, a = b = 0.472 nm, and c = 0.319 nm. As shown in Figure 10a, the unit cell included six atoms, including four oxygen atoms and two tin atoms. Theoretical studies have demonstrated that SnO_2_ has a direct bandgap near the Brillouin C-point zone. Figure 10b shows a typical diagram of the SnO_2_ band structure. Depending on the synthesis conditions, SnO_2_ bandgaps are stated in references as being between 3.6 and over 4.1 eV (even up to 4.5 eV at amorphous state) [30] All perovskite materials, including CsSnI_3_, FAPbI_3_, MAPbI_3_, and MAPbBr_3_, have a CB of approximately 3.4–3.9 eV, whereas the CB of SnO_2_ was 4.5 eV (Figure 7b). The high quality of the planar heterojunction was theoretically ensured by the high band alignment of SnO_2_ with the perovskite film. Furthermore, the high electron extraction of SnO_2_ aids in the efficient extraction of electrons from the perovskite. Studies have shown that SnO_2_/perovskite heterojunctions can have a higher built-in potential (0.94 V) than that of TiO_2_-based ETL, which delays the recombination of charge carriers and hence increases both the FF and V_oc_ [34]. The exceptional quality of the SnO_2_/perovskite heterojunction implies that electric parameters, such as RS and RSH, may be extracted from the internal resistance of the device using J–V curves.

SnO_2_ has an excellent transmittance of 91% in glass owing to its broad bandgap and low reflection index of 2 [35]. According to previous studies, SnO_2_ films can aid in boosting the transmittance of F-doped tin oxide (FTO) glass because they have a higher transmittance than that of TiO_2_ films (Figure 10c). With transmittances above 95% in the visible region, SnO_2_ QD films have been recently used as ETLs (Figure 10d). SnO_2_ can absorb less UV light while preserving device stability because it manages light in the UV-visible region better than TiO_2_, allowing a photon to readily pass through and be absorbed by the perovskite. Additionally, SnO_2_ has a deeper CB and 100 times greater electron mobility than that of TiO_2_.

During the preparation of ETLs, high temperature (HTP) is commonly required to hinder organics or additives. High temperature has a greater manufacturing cost and payback period in terms of energy. Only low temperature (LTP) is provided for flexible PSCs with a plastic substrate [37]. Consequently, enhancing the fabrication technique or inventing new routes should be adopted in the manufacture of LTP ETLs. TiO_2_ can also be fabricated using LTP. SnO_2_ ETLs have a much better scenario. The LTP is the preferred method for depositing SnO_2_ ETLs. One explanation is that annealing effects degrade HTP SnO_2_, resulting in poor interfacial contact and electrical characteristics. Additionally, the energy level is not matched with the nearby perovskite absorber. Furthermore, LTP SnO_2_ is a low-cost, simple-to-operate ETL material that has proven to be ideal for efficient PSCs. LTP is an excellent choice for fabricating SnO_2_ ETLs owing to its low cost, ease of production, and superior performance in PSCs.

The stability of PSCs is strongly influenced by their surroundings, which include the environment and contact layers of the perovskite absorbers. Regardless of the environment that the devices are stored in, the most popular contact layers of perovskite, such as TiO_2_ or ZnO, degrade significantly to 10–30% of their initial value after only a few hundred hours. The photic instability of TiO_2_ and the dissolution of perovskite absorbers generated by OH residual on the ZnO surface are the principal causes of such instability in TiO_2_ and ZnO PSCs, respectively. SnO_2_ has a relatively broad bandgap, which allows it to absorb less UV radiation while maintaining device stability. Additionally, the decreased acidity resistance and hygroscopicity of SnO_2_ provide the endurance to the PSC device. According to our research and several other independent studies, SnO_2_ PSCs offer high stability and potential in real-world applications.

Currently, over ten different procedures have been investigated for preparing SnO_2_ ETLs, including atomic layer deposition (ALD), sol–gel method, dual fuel combustion, electrodeposition, chemical bath deposition, hydrothermal method, e-beam evaporation, and ball milling.

### 3.2. Doped SnO_2_

It has been established that doping heteroatoms with Sn or O can result in an increase in carriers or holes. The preferred orientation, optical characteristics, and electrical characteristics of SnO_2_ film are enhanced by heteroatom doping.

The “population value (PV)” index is used to indicate bond formation before and after doping application. In general, strong ionic bonds are typically characterized by low PVs, whereas strong covalent bonds typically exhibit high PVs. The Sn–O bond’s total value falls after heteroatom doping, indicating that it contains more ions and exhibits strong iconicity, clear electron localization, and high electron affinity. The binding energy of the crystal lattice is given using the following equation, which can be used to assess the stability of the crystal structure of the doped lattice.
$$E_{bind}=(E_{\left(AB\right)}-E_{\left(A\right)}-E_{\left(B\right)})/n$$

E_(A)_ and E_(B)_ are the chemical potentials of atoms, E_(AB)_ is the overall energy of the doped structure, and n is the total number of atoms in the unit cell structure. The defect binding energy values of the doped systems are all negative, illustrating that all the doped crystal structures are stable structures, and conversely, the structures are unstable when the binding energy values are positive. 

The SnO_2_ electronic structure is altered as a result of doping along with the distortion of the crystal structure. In the bandgap of SnO_2_, the doping atoms form the impurity levels. When non-metal atoms are added, the Fermi level moves into the conduction band, revealing the metallicity of the SnO_2_ crystal. SnO_2_ has a low conductivity because the Fermi energy level is at the top of the valence band. The interaction between Sn 5s and O 2p orbitals, as well as the energy of Sn 5s orbits, are exaggerated, which causes a broader valence band and a narrower bandgap. The investigation of the electrical structure of SnO_2_ crystals is unaffected by variations in bandgap and energy band. It is important to note that doping drastically alters the energy band of SnO_2_. The SnO_2_ bandgap experiences doping orbitals, which raise the doped crystal’s Fermi level and increase the conductivity of the SnO_2_ crystal. When doping atoms are introduced, the conduction band of the SnO_2_ crystal is penetrated by the Fermi energy level, making SnO_2_ a conductor.

SnO_2_ is a popular transparent conductive film material that is used for doors, windows, and other conductive surfaces. It is also favored for coatings on low-emissivity glass. The infrared reflectivity of low-emissivity materials is their most significant characteristic. As a result, depending on the application, SnO_2_-doped materials are sometimes used to make low-emissivity glass coatings. SnO_2_ doping increases reflectance; when the light wavelength is greater than 1800 nm, the material exhibits iridescent reflection. When the wavelength is below 1800 nm, the material exhibits transparency.

In essence, the doping metal may cause metal ions at Sn^4+^ locations in the original SnO_2_ lattice to be substituted. More oxygen vacancies that behave as defects are introduced. Two benefits will result from the rise in these defects: (i) Bandgap energy of SnO_2_ will decrease when an appropriate metal dopant is used. The photocurrent of SnO_2_ layers will be enhanced by the decreased bandgap’s contribution to photogenerated carriers. (ii) The electrical conductivity of the SnO_2_ layer will likewise rise as the number of flaws increases. Within the SnO_2_ layer, it enhances electron transport. Because of the defects caused by oxygen vacancies in SnO_2_, doping a material with a metal content close to that of the material results in a reduction in optical bandgap. Therefore, photocarriers in the device would rise both quantitatively and be accelerated to the electrodes, increasing PCE. Nevertheless, a positive way to enhance the electrical conductivity of ETL is the effective rectification of defect concentrations (particularly oxygen vacancies in inorganic semiconductors) through doping with non-equivalent metal cations. Inorganic ETL’s electron trapping is generally limited by substitution defects of metal cations, particularly surface vacancies, which benefit PSC performance and stability. For instance, it has been demonstrated that eliminating oxygen defects through the use of a doping solution containing the proper amount of Nb^5+^ lowers the resistance, leading to high PCE and low J–V hysteresis [70]. Furthermore, Bai et al. used Sb^3+^-doped SnO_2_ as the electronic transfer layer (ETL) for PSC, which greatly increased the electronic conductivity [71]. Higher Fermi levels are also present in Sb^3+^-doped SnO_2_, which leads to better-tuned energy levels, a higher Voc, and less charge recombination. In contrast, the performance of a gadget is also adversely affected by a shortage of oxygen vacancies. In order to modify the oxygen vacancy density, Mg^2+^ was added to the porous SnO_2_ combination [72]. Mg^2+^ takes up both the Sn^4+^ site and the interstitial site of the ETL, increasing the number of oxygen vacancies. While intercalated Mg can make it easier for neutral oxygen vacancies to be ionized and produce electrons, the Mg lattice can promote the production of neutral oxygen vacancies, which is not favorable for electron density. As a result, the electron density considerably rises, and the Fermi level in the ETL is moved upward. In summary, solution-treated films after low-temperature annealing can produce high concentrations of oxygen vacancies, wherein the dopants act as a substitution for defects, namely oxygen vacancies; in contrast, ETLs (such as porous SnO_2_) with high-temperature sintering can cause oxygen defect density deficits, and the dopants play a role in complementing the defects under this condition. Figure 11 shows that the doping content has a significant impact on PCE performance. If the doping content is appropriate, it can improve PCE by reducing bandwidth, but if the concentration is high, recombination will increase. Thus, it is really necessary to determine the optimal concentration as well as the mechanism of introducing various types of doping.

## 4. Doping Engineering of SnO_2_

With favorable optoelectronic properties, SnO_2_ has been considered the most promising ETL material for PSCs. However, SnO_2_ faces the challenge of current–voltage (J–V) hysteresis, which significantly limits its application potential. Currently, surface engineering techniques have been employed to adjust energy levels, improve electron carrier characteristics, and contribute to efficiency in PSCs, in which doping techniques prove to be a superior solution to enhance ETL and electronic interface characteristics; enhancement promotes the transfer of electrons. While doping TiO_2_, the defect states and doping are highly dependent on the synthesis method, which makes it difficult to determine the suitability of the dopants. Thus, there is a rough prediction of dopant suitability via a specific synthesis method that involves doping with elements that have valences equal to that of the host ion TiO_2_. In the case of SnO_2_, the matching mechanism of the dopant was first determined using specific methods. Additionally, the factors determining the effectiveness of the dopants were examined for the first time. To date, four reviews have addressed the modification of SnO_2_ using doping techniques [1,73,74,75]. These reviews highlight some important aspects of doping-based SnO_2_. However, the descriptions are not sufficient to cover all aspects. Herein, various methods have been proposed for modifying the SnO_2_ surface. These methods can be categorized into four main approaches, including elemental doping, nonmetal doping, SnO_2_/composite structures, and organic doping.

The effectiveness of using a SnO_2_ ETL for PSC can be assessed using the most common parameters discussed in this review.

Conduction band: Because the open-circuit potential (V_oc_) is determined by the energy levels of the electron transport conduction band materials and the valence band of the hole transport material, doping-based SnO_2_ has a lower conduction band energy than that of pure SnO_2_, which has a conducive impact on device performance. Mott–Schottky analysis detected variations in conduction band energy.

Trap state: The trap state is formed via the reduction in oxygen in the SnO_2_ crystal lattice by combining electrons with similar ionic radii to Sn, thereby removing the oxygen vacancies and contributing to the enhancement of the electric transport pathway. The trap behavior is closely related to the doping content.

Recombination rate: The performance of the SnO_2_ ETL is primarily related to the carrier recombination rate related to the defect densities at the interface of the ETL/perovskite absorber, where the reduced recombination rate resulted in decreased shunt resistance and increased FF.

Charge transport rate: Electron transport is highly dependent on the current density, where a low current limits electron movement. By contrast, a high current will assist in reducing the trap state and increasing the number of free electrons, leading to improved PSC performance.

Grain boundary migration: The grain boundary alludes to the central region of charge recombination, where perovskites are more prone to decay, severely impeding the movement of charge carriers and causing a decrease in the efficiency of PSC. Therefore, fewer grain boundaries are necessary to reduce carrier coherence at the interface.

### 4.1. Metal Doping

The inclusion of metal nanoparticles into the SnO_2_ host substrate is the most widely used technique to address the issue of poor lifespan for electron–hole pairs in ETL. By lowering the doped SnO_2_’s absorption energy barrier, metal doping helps to increase photovoltaic efficiency. More significantly, the references to the doping effect highlight the propensity of the doping sites to serve as recombination hubs for electrons and holes, contributing to the longer lifetime of electron–holes pairs that are photoexcited, with the advantage of higher photovoltaics. Currently, various dopants are used in SnO_2_-based PSCs, such as zirconium, lithium, antimony, tantalum, niobium, gallium, and yttrium. The selection of doping elements is conducted in accordance with the rule that the selected ion should have an extra free electron and a similar ionic radius to that of Sn^4+^. Accordingly, Ta^5+^ was used as a doping material for SnO_2_. Owing to different condensation rates and hydrolysis of Ta^5+^ and Sn^4+^ ions, some changes in the size distribution of the Ta–SnO_2_ film caused an increase in the nucleation rate and the generation of smaller particles. Figure 12a depicts the optical transmission spectra of Ta-doped SnO_2_ and pristine SnO_2_ films coated on ITO, demonstrating improved transmittance in the 300–800 nm wavelength range. It was found that the as-prepared doped-SnO_2_ layer had greater transmittance, enabling the perovskite layer to absorb more light. In particular, Ta doping at 1.0% produced the highest conductivity and the best transmittance. It is evident from Figure 12b that the perovskite layer on Ta–SnO_2_ has higher absorbance than that of the pristine SnO_2_ layer. The improvement in absorption is most likely caused by the improved interface contact because of the indiscernible change between the shape of the perovskite layer deposited on pristine SnO_2_ and doped SnO_2_ ETLs. Furthermore, it is evident that the VTFL of the pristine SnO_2_ device (0.22 V) is greater than that of the doped SnO_2_ device (1.0%) (0.17 V) (as shown in Figure 12c,d. However, owing to the similarity in the ionic radii of Ta^5+^ (63 pm) and Sn^4+^ (68 pm), the crystal and nuclei still have a uniform dispersion without a significant change in the film properties, facilitating better contact with the perovskite absorbent layer, resulting in an enhancement in the PSC performance [76]. 

The same mechanism was observed with Ga^3+^ (62 pm), [77] Nb^5+^ (70 pm) [70,78,79,80], Cu^2+^ (72 pm) [81], Zn^2+^ (74 pm) [82], and Zr^4+^ (79 pm) [83]. Figure 13a–d display the effectiveness of doping when introduced into SnO_2_, resulting in superior performance compared to that of pristine SnO_2_. Obviously, because of less lattice distortion and similar metallic character, choosing doping elements with low variation in an ionic radius will result in the least change in elastic energy, which also facilitates their occurrence in the same chemical environment. 

The next chosen mechanism originates from the unique structure of rare earth elements. Owing to their partially full 4f orbitals and unoccupied 5d orbitals, rare earth metals have higher energy levels than any other metal. Lanthanide-related materials have highly intriguing spectral features, such as significant ferromagnetism and up-conversion luminescence, owing to the shielding of the 4f orbital using the fully filled 5p and 6s orbitals. Lanthanide doping is a useful method to adjust ETLs to produce distinct optical and electrical properties owing to its unique qualities [84]. The introduction of a small amount of La into the SnO_2_ lattice led to the observation of a pinhole-free morphological surface of SnO_2_ (Figure 14 b,d), in contrast to the pristine SnO_2_ growth on FTO (Figure 14a), which aggregated with the creation of additional pinholes (red circles). Pinholes and aggregated surfaces can significantly enhance recombination at the interface between the ETLs and the perovskite layer, which is a drawback for PSC applications. However, the morphological surface of the 5% La:SnO_2_ with pinholes was once more visible when the dopant concentration was increased, suggesting that optimizing the La concentration is crucial for the optimal PSC performance. It is evident that lanthanide doping significantly improves the SnO_2_ layer-covering quality and decreases the surface-state traps, resulting in a high capacity for collecting photogenerated electrons. Top-view SEM images of the perovskite layer deposited on the pristine SnO_2_ and La-doped SnO_2_ ETL are illustrated in Figure 14e,f. The consistent crystal size of approximately 400 nm and smooth surface covering was also observed in the perovskite film. Small crystals are also observed in the perovskite top SEM layer.

This element effectively passes the oxygen vacancy-related defects on the SnO_2_ surface, which reduces the interfacial defect density and inhibits charge recombination, further improving the quality of the perovskite films and the efficiency of PSCs. Notably, the doping concentration is strongly dependent on the ionic radii of the elements. Each element has its own optimum concentration to strike a balance between high charge carrier concentration and high mobility. For example, the best conductivity was achieved in SnO_2_ doped with 5 mol% Al^3+^ (53 pm) [86], whereas for Sb^3+^ (90 pm) [71] and Li^+^ (182 pm) [87], it was 4 mol% and 0.2 mol%, respectively. However, the concentration should not be exceedingly high because it will reduce the efficiency, which in turn will increase the effective activation energy of the donor and rapidly reduce the mobility of the carrier due to increased disorder or scattering. Rare earth elements contribute to the reduction of SnO_2_ crystal aggregation, leading to a pinhole-free morphology on the surface [85,88,89]. One factor that requires further mechanistic consideration in doping selection is the physical properties of the ETL. As a powerful interfacial modifier for preventing the development of Pb-halide antisite deep-trap defects, Cl ions have been extensively utilized to modify ETLs to reduce interface defects for effective PSCs [90]. It is possible that flaws on the ETL surface anchor grain boundaries on the wetting surface, preventing grain boundary migration. Consequently, the increased quality of the perovskite active layer was indicated by the hydrophobic SnO_2_-Cl ETL elicited by the presence of Cl [91]. 

### 4.2. Nonmetal Doping

An ideal ETL for high-performance PSCs should have high optical transmittance to allow sufficient light into the perovskite absorber, form energy levels that are comparable to that of perovskite to achieve a high open-circuit voltage (V_oc_) and generate high electron mobility to efficiently extract electrons from the perovskite to reduce charge recombination interfaces. In planar-type PSCs, fast charge carriers are required to efficiently limit charge accumulation from ion migration at the interface, resulting in reduced hysteresis. However, it is widely recognized that the morphology of the numerous interfaces in PSCs affects the charge carrier transport efficiency, making it critical to adjust the interface morphology of ETL/perovskite to promote electron transportation in order to achieve the desired device performance. The morphology and crystallinity of perovskite crystal films have been shown to improve as a result of the widespread use of carbon derivatives, such as carbon nanotubes, graphene, fullerene, and graphene oxide, as additives or interlayers in PSCs. They have also been shown to provide effective charge-collection channels and facilitate electron extraction.

Consequently, it is worth investigating the use of such carbon derivatives as additives to change SnO_2_ to make a better ETL for PSCs. In this section, the mechanism of non-metal doping selection is similar to that of organic doping involving functional groups that exist on the surface of the material. Based on this mechanism, hydroxyl- and carboxylic-acid-rich red-carbon quantum dots (RCQs) were used to synthesize a doped SnO_2_ composite ETL (SnO_2_-RCQs) [62]. Interestingly, after RCQ doping, the SnO_2_ surface became more hydrophilic, indicating a reduction in the Gibbs free energy surface, which, as previously indicated, favors heterogeneous perovskite nucleation and produces high-quality perovskite films. On the surfaces of the SnO_2_-RCQs, hydroxyl- and carboxylic-acid-rich RCQs can improve the perovskite crystallinity, and as a result, the grain size increased. In addition to improving the perovskite film crystallization, doping RCQs into SnO_2_ also improves the microstructural uniformity of the film, which is desirable for large PSCs with superior performance. The SnO_2_-RCQs and SnO_2_ films were prepared using the same one-step solution method as the perovskite films. The SEM images of the perovskite films in Figure 15a demonstrate the change in the perovskite film morphology caused by the RCQ-doping of the SnO_2_ film. It is evident that both films are continuous, free of pinholes, and included randomly connected grains. However, compared with SnO_2_, the majority of the grains on the SnO_2_-RCQs are noticeably larger. As a result, the abundant RCQs in hydroxyl and carboxylic acids on the surface of SnO_2_-RCQs trigger perovskite crystallinity and, consequently, an increase in grain size. Further evidence of enhanced photogenerated carrier generation with reduced recombination is evident in the perovskite film grown on SnO_2_-RCQs, which also exhibits higher optical absorption (Figure 15b) and weaker photoluminescence (PL) emission (Figure 15c) than that of the control film. This enhanced photogenerated carrier generation with reduced recombination will eventually result in increased photocurrents in PSCs. The enhanced extraction of electrons in the perovskite film grown on SnO_2_-RCQs was further supported by time-resolved photoluminescence (TRPL) measurements (Figure 15d), which revealed that the lifetime of an electron in this film was significantly reduced from 103 ns in the control film grown on SnO_2_ to 44 ns. Moreover, 2D-GIXRD measurements were carried out based on the synchrotron to further demonstrate the shift in film crystallization on the SnO_2_-RCQs film and 2D-GIXRD measurements. Figure 15e illustrates that the diffraction profiles of the perovskite films on SnO_2_-RCQs and SnO_2_ are comparable to those of the textured, highly crystalline perovskite films. In Figure 15f, we provide the q-dependent 1D-GIXRD spectra around the perovskite (110) peak in Figure 15e to evaluate the crystallinity difference between both films.

In other words, the hydrophilicity, shape, and chemical groups connected to the ETL directly affect the perovskite layer’s quality. The non-metals (such as N, F, and P) are also employed in PSCs, which have improved performance as a result of non-metal doping [92,93,94]. These dopings can result in an electron transport surface that is smooth and hydrophobic, ensuring perovskite films with high crystallinity and big grains. With fewer nucleus nucleation sites and more spacing, the non-wetting and smooth ETL produces larger grains with fewer grain borders. Consequently, the increased nucleus separation allows for the growth of massive perovskite grains. This can be explained by the ETL surface’s smoothness and lack of wetness, which inhibit heterogeneous nucleation and promote grain boundary movement. This data support the idea that perovskite can grow its grain sizes because of its hydrophobic surface. Large grain sizes have fewer flaws at the grain boundaries, which lowers the amount of charge non-radiative recombination at the perovskite/ETL interface. This is primarily attributable to the non-metal-doped SnO_2_ and perovskite’s favorable interface contact and energy level alignment, which minimize the density of interface defects, speed up the extraction of electrons, and lessen energy loss from nonradiative recombination. In general, a hydrophobic surface will cause the perovskites’ grains to grow, which will boost PSC performance.

### 4.3. Composite Doping

It is believed that the condition of the substrate interface, type of defect, and makeup of the film are related to the quality of the ETL and perovskite [95]. Examining the interface between the ETL and perovskite layers of the device, researchers have found complex changes, such as dipole generation and band bending. A good film quality, good interface contact with the perovskite layer, and energetic alignment are essential for the ETL. To achieve these goals, several passivation methods for ETL have been implemented. The mechanism of composite doping is based on the synergistic effects of the ingredients. The halogen atoms have the smallest radius, which means that they can be easily moved, resulting in high applicability for the PSC-printed passivation surface. Additionally, halides added to perovskite precursor solutions limit the trap density and reduce non-radiative recombination [12,96,97]. Moreover, it has been demonstrated that alkali metals have a positive effect on defect passivation in perovskite layers and grain boundaries. The presence of K^+^ promoted a change in the perovskite structure with horizontal grain boundaries, which facilitated carrier diffusion, resulting in superior photoelectric efficiency. However, with other alkali metal elements, Na^+^ contributed to growing perovskite thin films by increasing the particle size to enhance device performance [98,99]. Therefore, the synergistic effect between the alkali metal and halide improves the performance of the device compared with that of pure SnO_2_. Indeed, NaCl-, KCl-, and KF-doped SnO_2_ optimize the perovskite morphology, forming a homogeneous layer that leads to increased electrical conductivity and reduced defect density in the perovskite film, thereby enhancing the electric transfer efficiency from the perovskite film to the ETL [100,101,102]. Based on this inspiration, SbCl_3_ is also utilized as a doping material in SnO_2_ ETL based on its ability to increase the charge-carrier concentration, high mobility of Sb^3+^, and high conduction band of Zr [103,104]. It should be noted that different doping methods improve the properties and electrical conductivity of SnO_2_ films in different ways, such as by promoting the separation of SnO_2_ colloidal particles to form a compressed layer that increases electron density, provides freedom from doping materials, and reduces defects in ETLs. The existence of doped composites increases the density of the SnO_2_ layer, leading to the formation of additional electron transport pathways, thereby improving the PSC efficiency. In addition, inorganic metal oxides are also used as doping materials in SnO_2_ ETLs. The combination with metal oxides increased the thickness of the SnO_2_ layer, leading to energy leakage caused by the Joule effect. Therefore, a suitable doping selection mechanism is based on the high-impedance nature of the material. Based on this condition, PbO, Al_2_O_3_, and ZnO were utilized as a potential inorganic metal-oxide-doped SnO_2_ ETL in PSCs, forming a double ETL [105,106,107]. Reference samples with bare FTO and FTO/SnO_2_ are shown in the top-view SEM images in Figure 16a–f [105]. It is evident that SnO_2_ evenly covered the FTO and began to aggregate. By contrast, when Pb was present, the thin film was altered, and the amount of vital particle aggregates increased as the lead acetate concentration in the precursor solution increased. The strong nanoparticle aggregation prevented the FTO film from being completely coated at the highest Pb content of 6%. The targeted double ETL enhanced the shunt resistance of the device, thus avoiding the leakage of the photogenerated current. In addition, the introduction of metal ions into the crystal lattice increased the distance between crystals because of the replacement of some Sn atoms by larger Pb atoms, which in turn increased electron diffusion pathways and contributed to PSC performance enhancement. Meanwhile, SnO_2_–PbO demonstrated that depolarization occurred much more slowly between 500 and 1000 s. This result indicates that the device depolarizes owing to a delayed leakage of charge caused by a decreased shunt resistance (R_sh_) associated with the parallel current channel, as detailed below. SnO_2_–PbO devices gain from high R_sh_, whereas SnO_2_ devices suffer from low R_sh_ (increased parallel pathways). The charge carriers were evaluated using the space charge-limited current (SCLC) process, as illustrated in Figure 16, to determine the impact of charge transmission within the ETL. The relatively better capacity of SnO_2_-PbO to absorb electrons is confirmed by its shorter photoluminescence (PL) lifespan and considerable PL quenching; however, these factors do not completely explain the significant difference in V_oc_ observed under low illumination conditions. The single-diode model can be used to determine the shunt resistance from the J–V measurements performed in the dark (Figure 16i) [28] These variables provide clear details of parallel charge channels that are not involved in the photocurrent. With values of 22.2 and 671.2 k cm^2^ for SnO_2_ and SnO_2_–PbO, respectively, the results clearly demonstrate variations in R_sh_ of more than one order of magnitude.

### 4.4. Organic Doping

The utilization of organic doping materials in SnO_2_ ETLs is related to the term “work function (WF)”. The WF is a concept that covers the interface area between the ETL and perovskite, as well as the organic-doped SnO_2_ ETL. At the ETL/perovskite interface, the WF exhibits a measurable difference in energy levels between the conduction band of SnO_2_ and the valence band of perovskite, while the WF of the organic-doped ETL refers to functional groups existing in the complex structure. Unlike elemental and composite doping with the typical physical properties of metals, the role of functional groups in organic doping affects the quality of perovskite films. Polyethylenimine (PEIE) possesses a high density of amine groups, where the electron-donating ability of tertiary amine groups presents a high density of amine groups, resulting in enhanced extraction capabilities, charge transport, and reduced charge recombination, all of which improve device performance. Another organic polymer was used (PVP and PEG) based on hydrogen bonding between the organic polymer and the SnO_2_ ETL, which hindered the agglomeration of the nanoparticles. Additionally, perylene diimide (PDI) derivatives have attracted considerable research interest because of their excellent stability and ability to facilitate charge-carrier transportation by modulating the interlayer working function. Therefore, the mechanism for selecting organic doping based on the functional groups existing on the surface of the material can effectively interact with the SnO_2_ ETL, which reduces WB. Moreover, it is conducive to creating high-quality pinhole-free perovskite films, contributing to improved charge transportation and reduced charge recombination. Figure 17a illustrates a schematic of the PEG-incorporated SnO_2_ ink, where the PEG molecules act as ligands around the SnO_2_ nanoparticles. This behavior suggests that the SnO_2_ oligomers can disaggregate because of the strength of the hydrogen bonding between the PEG and SnO_2_ particles. Additionally, we discovered that the SnO_2_ particles in PEG8000 were tightly regulated below 20 nm, which is the optimal size for homogenous and sufficiently thin SnO_2_ films. The spin-coated compactness and homogeneity of the film are expected to enhance with PEG-incorporated SnO_2_ ink for two reasons (Figure 17b). To decrease the pinholes for films thinner than 30 nm, it is first necessary to prevent the formation of large ink clusters. Second, the PEG polymer can create a compact and dense matrix on the substrate. The roughness of pure SnO_2_ was compared with that of SPM on an ITO substrate using atomic force microscopy (AFM) to assess the film quality (Figure 17c,d). The SPM film had significantly less roughness (RMS:5 nm) than that of pure SnO_2_ film without the polymer (RMS:25 nm). Moreover, traditional SnO_2_ films require plasma/UV pretreatment to fully cover the substrate because it is an unwettable surface for the perovskite precursor solution. Fortunately, the insertion of PEG into the SnO_2_ film can enhance the affinity between SnO_2_ and perovskite by altering the SnO_2_ surface, as demonstrated in Figure 17e,f. On the pristine SnO_2_ film and SPM, the perovskite solution had contact angles of 52.9° and 11°, respectively. As a result, the perovskite coating on the SPM layer was completely covered without any pretreatment (such as plasma treatment). The crystal grains of the perovskite films on SPM were identical to those on the SnO_2_ substrates, according to the top-view SEM images (Figure 17g,h). Subsequently, we constructed planar solar cells based on the ITO/SPM to test the superior properties of the SPM film. ITO/SPM (or SnO_2_) ETL/perovskite/spiro-OMeTAD/Au comprises the entire device structure. The films were coated with spiro-OMeTAD and Au. Each layer clearly defined the border points for the presence of thick coatings on the substrate (Figure 17i).

The doping elements used for the SnO_2_ ETL with the main contributions to improving the properties of SnO_2_ are summarized in Table 1. SnO_2_ doping improves PSC performance, which is not trivial.

In addition to the aforementioned SnO_2_ ETL, other inorganic materials, such as binary and ternary metal oxides are also employed as ETLs in PSCs. Detailed information is summarized in Table 2. The best nitrogen-doped, SnO_2_-based PSC with PCE of 23.41%, significantly higher than other oxide PSCs, demonstrates the outstanding potential of doping engineering, opening up new opportunities to further develop the performance of metal oxide ETLs in PSCs.

## 5. Outlook and Perspective

In recent years, the PCE has increased from 3.76% to over 22.6%, which has significantly improved the performance of PSCs using various ETL. In this article, we provide a brief overview of recent PSC research conducted using a SnO_2_ ETL. Low-temperature approaches are necessary for flexible PSC devices; therefore, SnO_2_ nanoparticles have become a potential candidate for these applications owing to their low temperature, wide range of fabrication techniques, and the ability to offer a clear advantage over TiO_2_ ETL materials. The ability to precisely regulate SnO_2_ ETL features, such as the film quality, flaws, and surface work function, will be necessary for the future to increase the charge transfer efficiency by lowering interface recombination and energy loss. The objective is to create PSC devices based on SnO_2_ that are as stable and effective as devices with a mesoporous structure. This discrepancy is primarily owing to the higher density of surface flaws in SnO_2_ than that in TiO_2_. Reduced VOC and FF are caused by electron recombination at the perovskite/SnO_2_ interface, which is mediated by intrinsic and extrinsic surface defects that result in acceptor or donor states inside the band gap. Charge accumulation at the perovskite/SnO_2_ interface is one of the main sources of hysteresis, which is also caused by the increased density of surface defects. Weber et al. demonstrated that current-voltage hysteresis in PSCs is dominated by the dynamic creation and release of ionic charges at the interfaces. Numerous researchers have claimed that hysteresis can be eliminated using techniques such as interface modification, doping in SnO_2_ or perovskite, and mesoporous structures.

To further improve the V_OC_ values and fill factors of PSC devices, extensive work is required to eliminate non-radiative recombination and enhance charge transfer. Numerous methods, including changing the interface, doping the ETL and perovskite layer, and enlarging the perovskite grain size, have been reprinted and used to enhance the performance of PSC and eliminate hysteresis. TiO_2_ will continue to compete with SnO_2_ ETL-based PSC research in the future. Extensive studies aimed at enhancing charge collection and lowering interface recombination will improve the quality of the SnO_2_ ETL, which will result in more effective charge transfer. The lack of long-term stability of SnO_2_ ETL-based PSCs is a significant problem that must be urgently resolved. Interdisciplinary research is required to discover novel stable materials, better electrode options, barrier layers, charge transport layers, and encapsulation techniques for improving the stability of PSCs.

## Figures and Tables

**Figure 1 materials-16-06170-f001:**
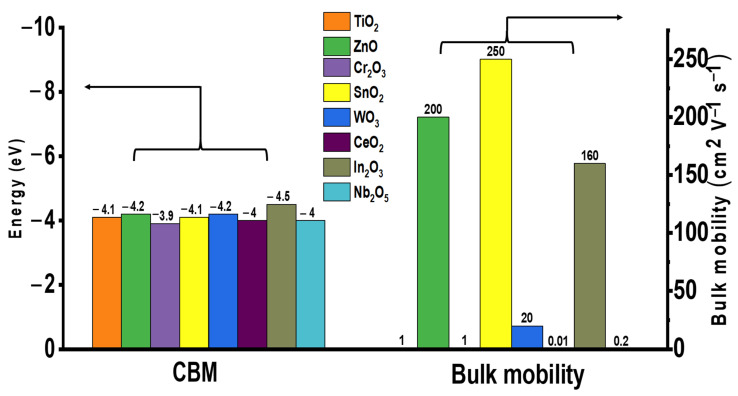
Electronic characteristics of metal oxide ETL for PSCs.

**Figure 2 materials-16-06170-f002:**
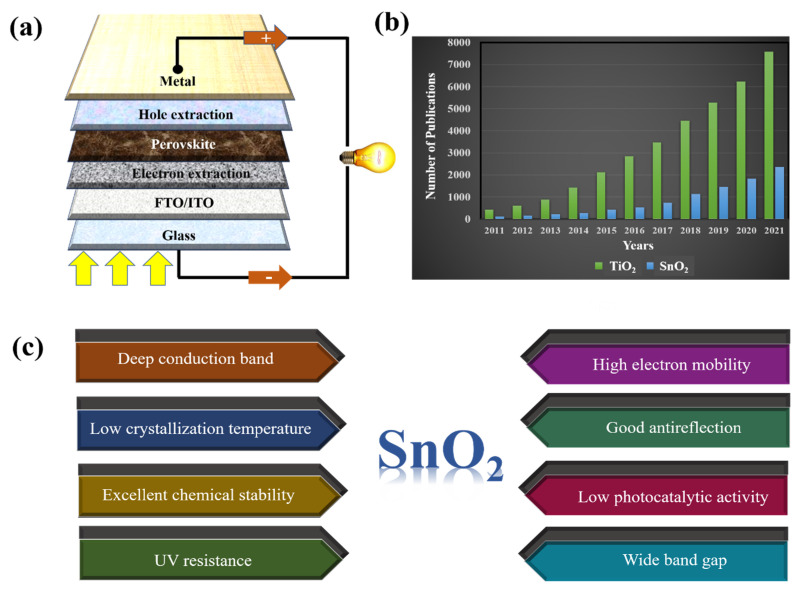
(**a**) A typical framework of PSCs. (**b**) The number of publications on TiO_2_ and SnO_2_ ETLs in PSCs from 2011 to 2021 (sourced from Google Scholar; search time: 20 June 2022). (**c**) Advantages of SnO_2_ when used as ETL in PSCs.

**Figure 3 materials-16-06170-f003:**
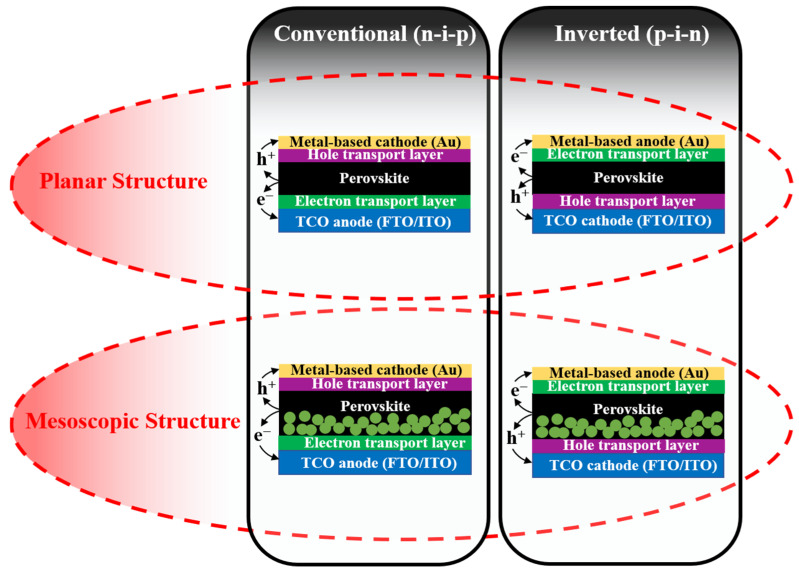
Schematic for configurations of mesoscopic and planar perovskite solar cells with either conventional (n-i-p) or inverted (p-i-n) structure.

**Figure 4 materials-16-06170-f004:**
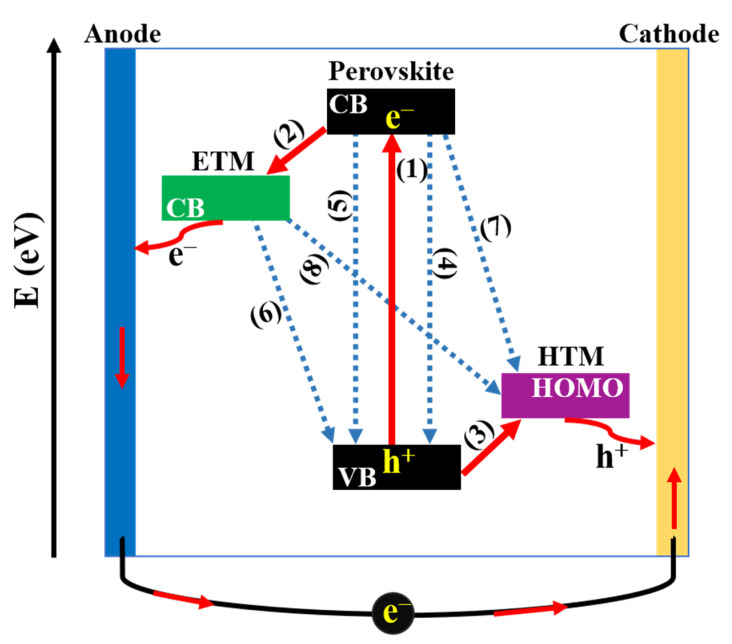
Schematic representation of charge-transfer processes in perovskite solar cells.

**Figure 5 materials-16-06170-f005:**
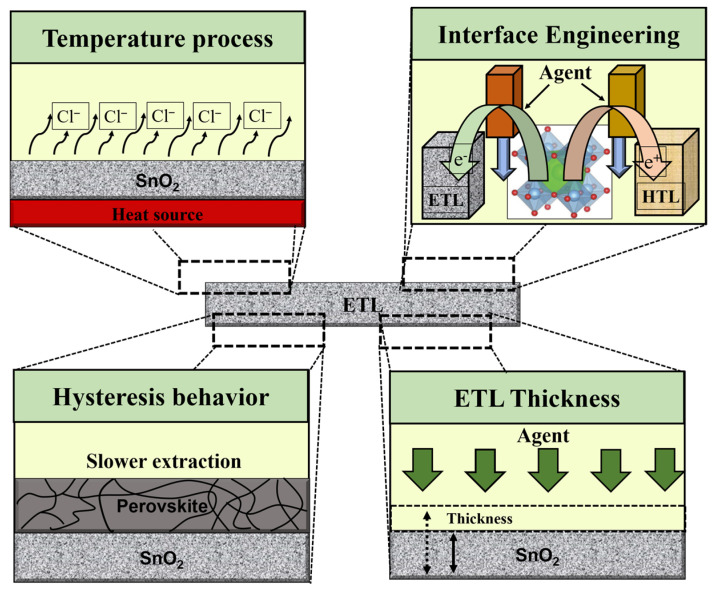
Illustration of the main challenges of SnO_2_ ETL observed in PSCs.

**Figure 6 materials-16-06170-f006:**
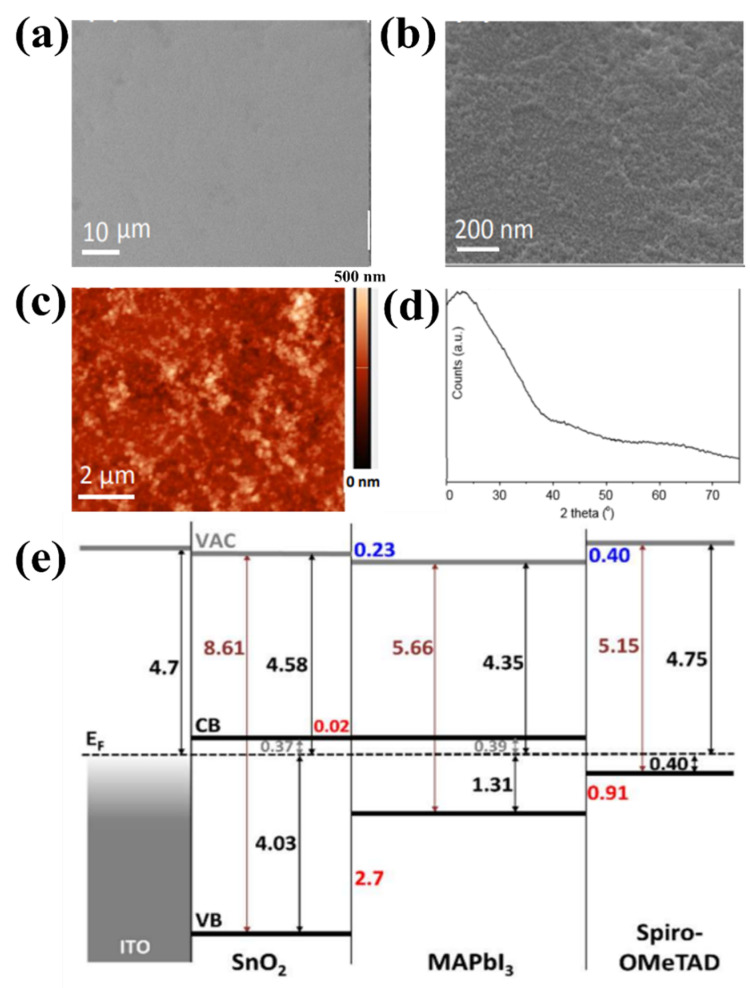
SEM image at (**a**) low and (**b**) high magnification of as-prepared SnO_2_ on glass/ITO substrate. (**c**) AFM scan of as-prepared SnO2. (**d**) XRD spectrum of as-prepared SnO_2_ on glass. (**e**) Energy level diagram of the ITO/as-prepared SnO_2_/perovskite/Spiro-OMeTAD layer stack determined using UPS and UV–Vis spectrophotometry. Reproduced with permission from Barbe et al. [50]. Copyright 2017, American Chemical Society.

**Figure 7 materials-16-06170-f007:**
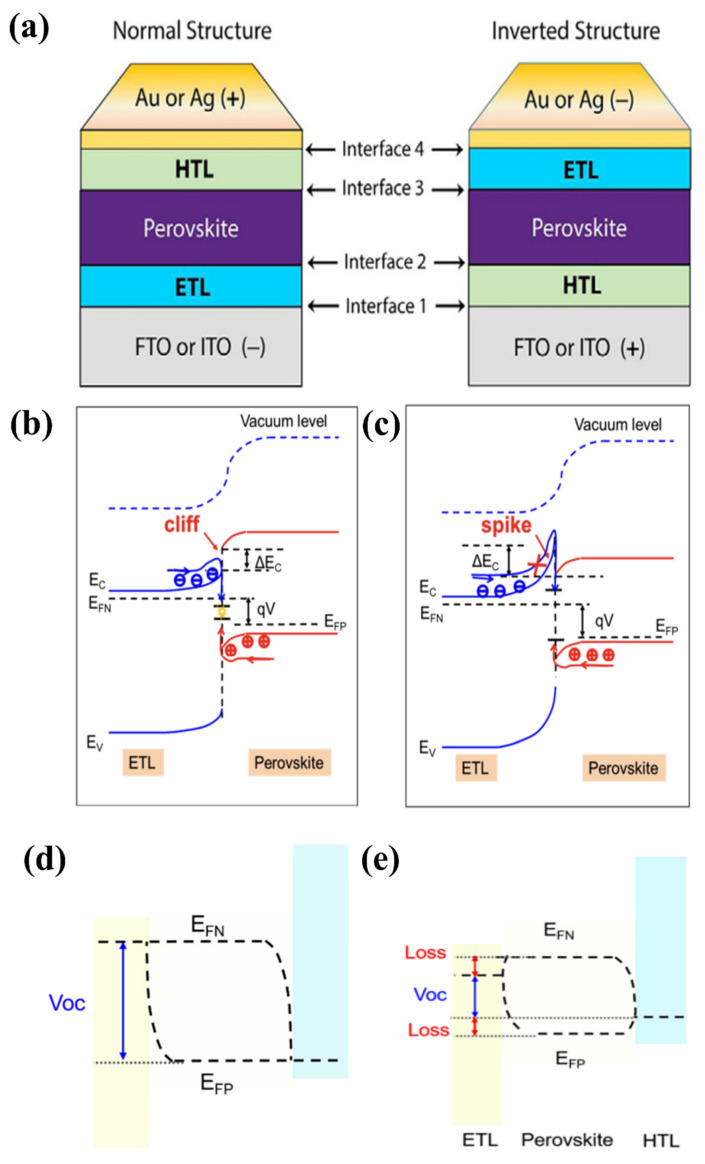
(**a**) Normal and inverted structures of PSC. Reproduced with permission from Cho et al. [52]. Copyright 2017 Wiley-VCH Verlag GmbH & Co. KGaA, Weinheim, Germany. ETL/perovskite forward-biased energy band plot at (**b**) cliff structure and (**c**) spike structure, where E_F_ indicates Fermi energy level, E_V_ indicates maximum valence energy, ΔE_c_ represents degree conduction band deviation, and red circles represent holes and blue circles represent electrons. E_F_ and V_oc_ in the PSC are depicted schematically for illumination with (**d**) well-aligned bands and (**e**) significant energy offset between the transport layer (ETL or HTL) and the perovskite layer. Reproduced with permission from Pan et al. [53]. Copyright 2021 AIP Publishing.

**Figure 8 materials-16-06170-f008:**
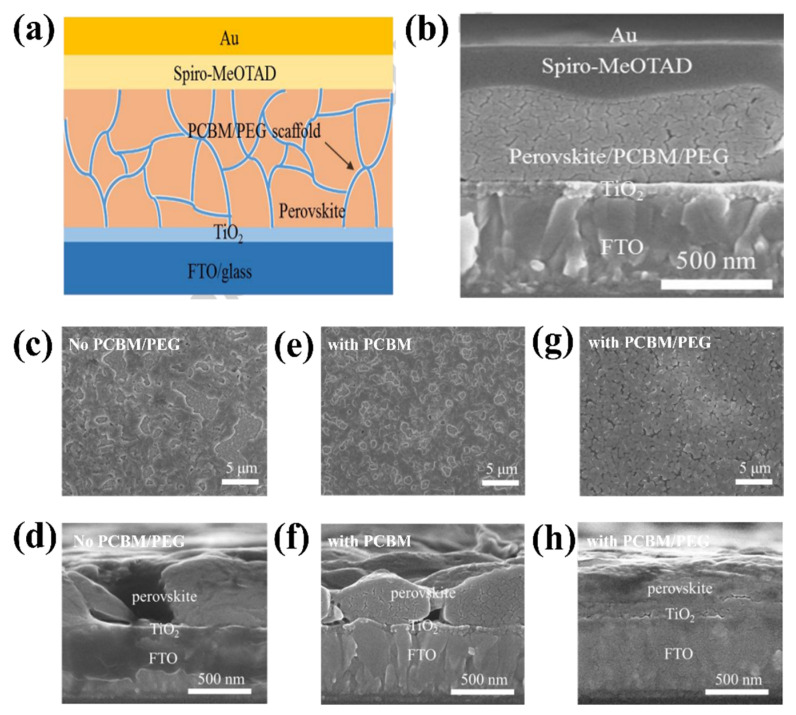
(**a**) Structure of PSC with PCBM-PEG network. (**b**) Cross-section SEM image of PSC device based on PCBM/PEG network. SEM images of perovskite film with or without PCBM/PEG. (**c**,**e**,**g**) Top-view SEM images of Film 1, 2, and 3. (**d**,**f**,**h**) Cross-section SEM images of Film 1, 2, and 3. All the films are coated on the TiO_2_/FTO/glass substrate. Reproduced with permission from Wei et al. [55]. Copyright 2016 Elsevier Ltd.

**Figure 9 materials-16-06170-f009:**
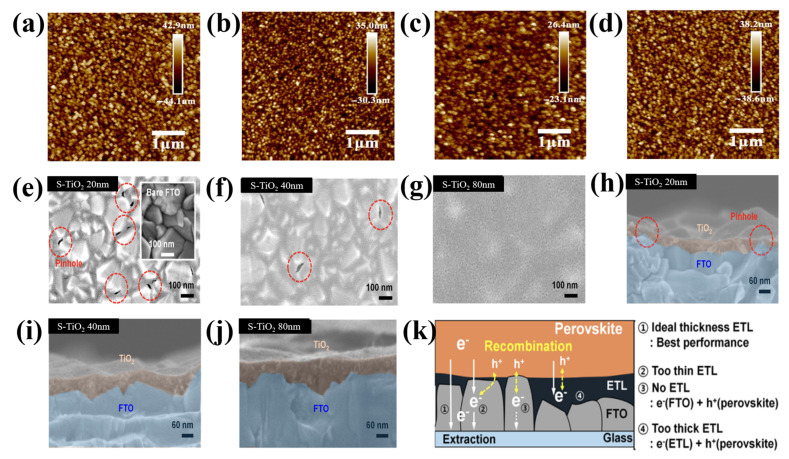
AFM images of (**a**) bare FTO, (**b**) c-TBOT, (**c**) c-TTIP, and (**d**) c-TTDB. Reproduced with permission from Qin et al. [64]. High-resolution scanning electron microscopy images of (**e**–**g**) the top and the corresponding (**h**–**j**) cross sections of S-TiO_2_ on FTO glass. (**k**) Schematic of the predicted electron collection process in the S-TiO_2_ for various thicknesses. Reproduced with permission from Choi et al. [65]. Copyright 2016 American Chemical Society.

**Figure 10 materials-16-06170-f010:**
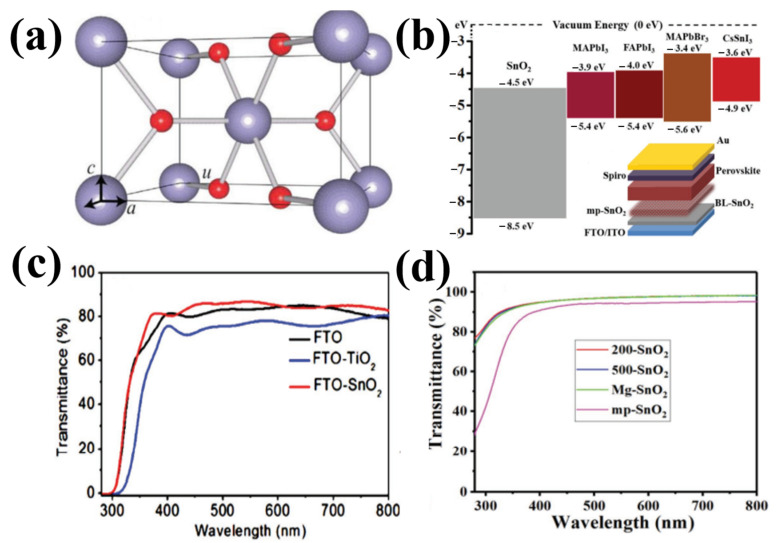
(**a**) Bonding structure of rutile SnO_2_. (**b**) Diagram illustrating the heterojunction SnO_2_/energy perovskite’s levels (relative to the vacuum level). (**c**) Transmission spectrum of SnO_2_ films deposited on FTO substrates. (**d**) Transmission spectra of SnO_2_ films deposited on a silica glass substrate. Reproduced with permission from Fang et al. [69]. Copyright 2018 WILEY-VCH Verlag GmbH & Co. KGaA, Weinheim, Germany.

**Figure 11 materials-16-06170-f011:**
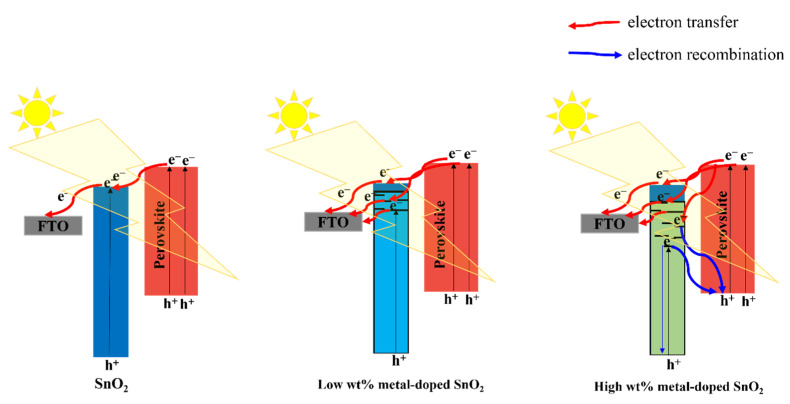
Schematic illustration of the impact of different doping concentrations on the SnO_2_ properties.

**Figure 12 materials-16-06170-f012:**
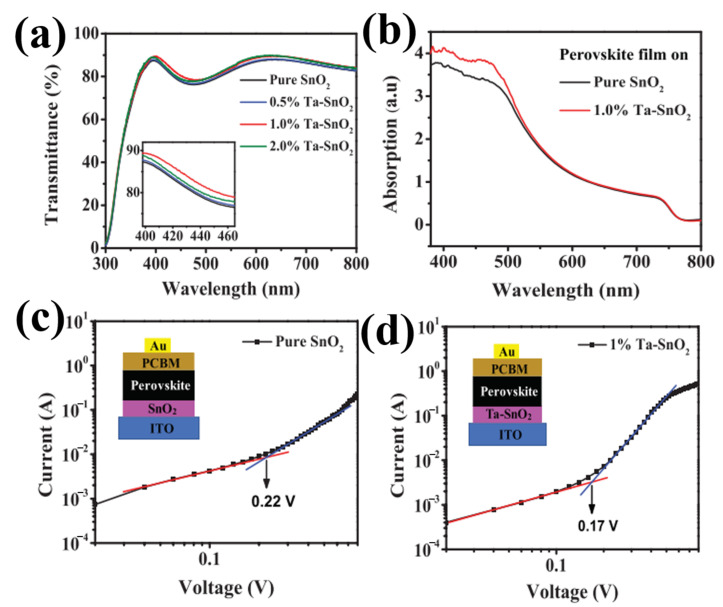
(**a**) Transmittance of pristine SnO+ and Ta–SnO+ films with different Ta contents deposited onto ITO substrates. (**b**) UV-visible absorption spectra of the perovskite films on pristine SnO_2_ and 1.0% Ta–SnO_2_. The dark J–V curves of the electron-only devices with (**c**) pristine SnO_2_ and (**d**) 1.0% Ta–SnO_2_ ETLs (red and blue line represent the fitting line). Reproduced with permission from Liu et al. [76]. Copyright 2019 AIP Publishing.

**Figure 13 materials-16-06170-f013:**
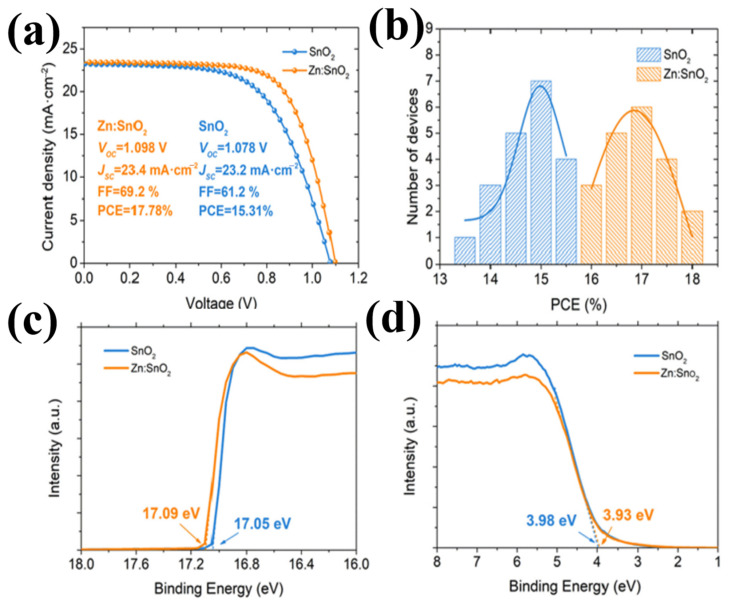
(**a**) J–V curves of the best performing PSC with SnO_2_ or Zn:SnO_2_ as ETL. (**b**) Histogram of PCEs for 20 devices using Zn:SnO_2_ or SnO_2_ as ETLs. (**c**,**d**) UPS spectra exhibiting the E cut-off and EF edge with different ETLs. Reproduced with permission from Ye et al. [82]. Copyright 2019 Elsevier B.V.

**Figure 14 materials-16-06170-f014:**
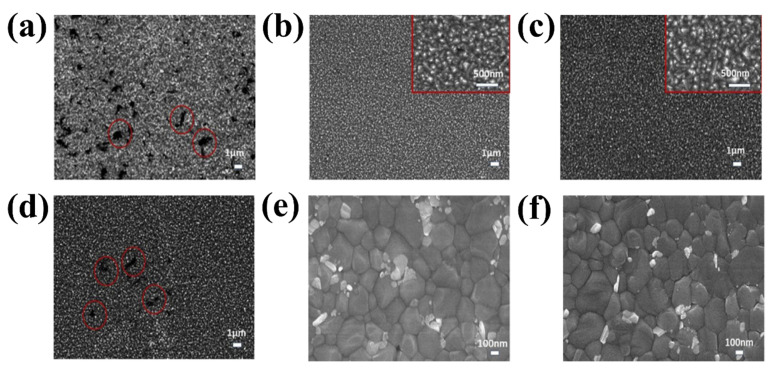
SEM image of (**a**) pristine SnO_2_, (**b**) 1% La:SnO_2_, (**c**) 2.5% La:SnO_2_, and (**d**) 5% La:SnO_2_ layer. SEM image of perovskite crystal growth on (**e**) pristine SnO_2_ and (**f**) La:SnO_2_. Reproduced with permission from Xu et al. [85]. Copyright 2019 Elsevier B.V.

**Figure 15 materials-16-06170-f015:**
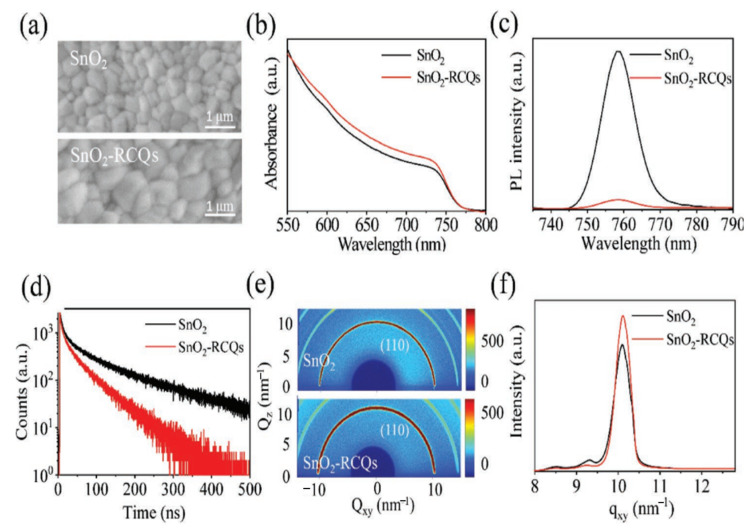
(**a**) SEM images, (**b**) optical absorption spectra, (**c**) PL spectra, (**d**) TRPL spectra, (**e**) 2D-GIXRD patterns, and (**f**) derived 1D-GIXRD spectra of the perovskite films grown on SnO_2_-RCQs and SnO_2_. Reproduced with permission from Huang et al. [62]. Copyright 2019 WILEY-VCH Verlag GmbH & Co. KGaA, Weinheim, Germany.

**Figure 16 materials-16-06170-f016:**
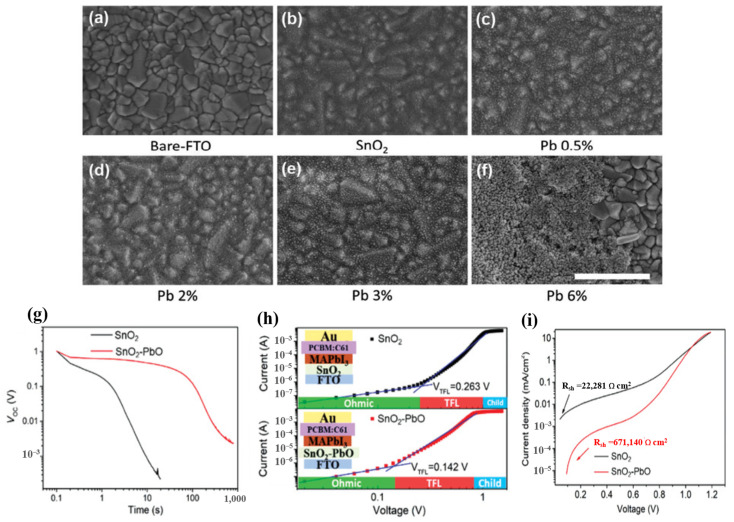
(**a**–**f**) Top-view SEM images of the SnO_2_ with different Pb doping; the scale bar is 500 nm. (**g**) Open-circuit voltage decay of SnO_2_ and SnO_2_–PbO ETLs based on PSCs. (**h**) Dark J–V curves of electron-only device with a configuration of FTO/SnO_2_ or SnO_2_–PbO ETLs/perovskite/PCBM/Au. (**i**) J–V curves measured in the dark and extraction of R_sh_. Reproduced with permission from Xu et al. [105]. Copyright 2021 Wiley-VCH GmbH.

**Figure 17 materials-16-06170-f017:**
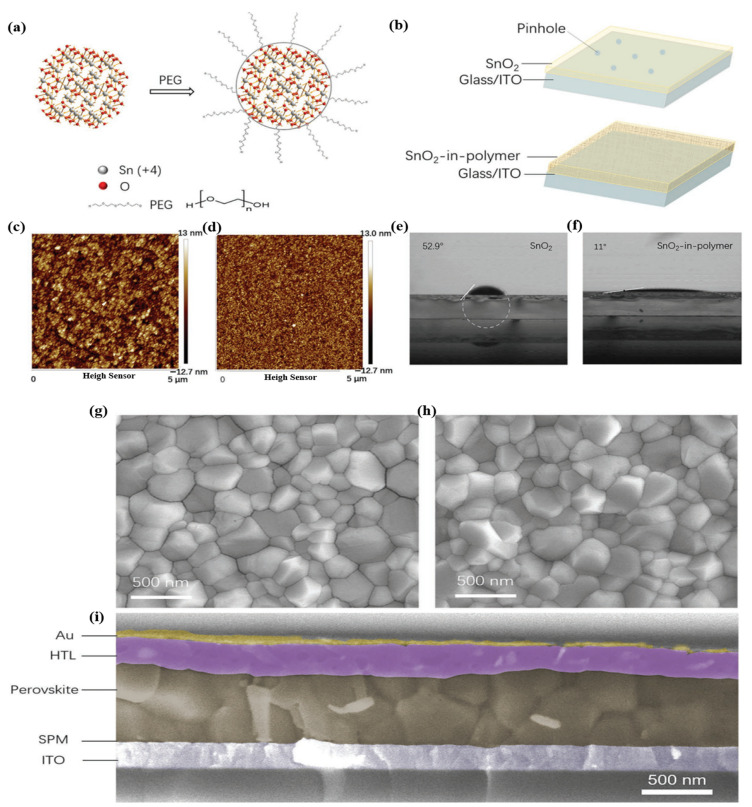
Dispersity of SnO_2_ ink with and without PEG. (**a**) Schematic of the interaction between PEG and SnO_2_. (**b**) Schematic of the film morphology for SnO_2_ and SPM. (**c**,**d**) AFM image and (**e**,**f**) contact angle measurement by fitting a white circle and dotted line to the interface of SnO_2_ films with (**d**,**f**) or without (**c,e**) PEG. (**g**,**h**) SEM images of perovskite films on SnO_2_ (**g**) and SPM (**h**) substrates. (**i**) Cross-section SEM image of the SPM-PSC device structure. Reproduced with permission from Xu et al. [108]. Copyright 2018 WILEY-VCH Verlag GmbH & Co. KgaA, Weinheim, Germany.

**Table 1 materials-16-06170-t001:** Dopants and their contribution to the improvement of PSCs based on SnO_2_ ETL.

Device Structure	J_sc_ (mA cm^−2^)	V_oc_ (V)	FF	PCE (%)	Ref.
FTO/SnO_2_/Cs_x_(MA_0.17_FA_0.83_)_(100−x)_Pb(I_0.83_Br_0.17_)_3_/CuPc/Carbon	23.2	1.078	0.612	15.31	
Zn-doped	23.4	1.098	0.692	17.78	[82]
AZO/SnO_2_/Cs_x_(MA_0.17_FA_0.83_)_(100−x)_Pb(I_0.83_Br_0.17_)_3_/Spiro-OMeTAD/Au	22.1	0.997	0.570	12.5	
Ga-doped	22.8	1.070	0.700	17.00	[77]
ITO/SnO_2_/Cs_x_(MA_0.17_FA_0.83_)_(100−x)_Pb(I_0.83_Br_0.17_)_3_/Spiro-OMeTAD/Au	21.7	1.158	0.777	19.48	
Ta-doped	22.8	1.161	0.786	20.80	[76]
FTO/SnO_2_/TiO_2_-MAPbI_3_-ZrO_2_/Carbon	22.8	0.880	0.610	12.32	
Nb-doped	24.1	0.920	0.610	13.53	[79]
ITO/SnO_2_/Cs_x_(MA_0.17_FA_0.83_)_(100−x)_Pb(I_0.83_Br_0.17_)_3_/Spiro-OMeTAD/Au	22.4	1.114	0.718	18.05	
Nb-doped	22.9	1.125	0.736	18.92	[70]
FTO/SnO_2_/CsFAMA/Spiro-OMeTAD/Au	22.7	1.130	0.727	18.64	
Nb-doped	23.2	1.100	0.786	20.07	[80]
FTO/SnO_2_/CsPBBr_3_/Carbon	7.5	1.270	0.648	6.73	
Nb-doped	8.9	1.290	0.695	8.54	[78]
FTO/SnO_2_/CsFAMA/Spiro-OMeTAD/Ag	23.2	1.078	0.771	19.43	
Cu-doped	24.2	1.108	0.790	21.35	[81]
ITO/SnO_2_/(FAPbI_3_)_0.95_(MAPbBr_3_)_0.05_/Spiro-OMeTAD/Au	24.3	1.060	0.660	17.30	
Zr-doped	24.7	1.080	0.720	19.54	[83]
ITO/SnO_2_/(FAPbI_3_)_0.95_(MAPbBr_3_)_0.05_/Spiro-OMeTAD/Au	22.8	0.993	0.780	17.43	
Gd-doped	23.8	1.143	0.820	22.40	[89]
FTO/SnO_2_/MAPbI_3_/Spiro-OMeTAD/Au	20.7	1.060	0.650	14.24	
La-doped	21.8	1.090	0.720	17.08	[85]
FTO/SnO_2_/CH_3_NH_3_PbI_3_/Spiro-OMeTAD/Au	18.6	1.030	0.610	11.69	
Y-doped	21.8	1.070	0.670	15.60	[88]
FTO/SnO_2_/(FAPbI_3_)_0.85_(MAPbBr_3_)_0.15_/Spiro-OMeTAD/Au	21.0	1.020	0.59	15.07	
Cl-doped	23.0	1.110	0.69	18.10	[91]
FTO/SnO_2_/MAPbI_3_/Spiro-OMeTAD/Au	16.8	1.000	0.530	9.02	
Al-doped	19.4	1.030	0.580	12.10	[86]
ITO/SnO_2_/MAPbI_3_/Spiro-OMeTAD/Au	22.3	1.010	0.696	15.70	
Sb-doped	22.6	1.060	0.742	17.70	[71]
FTO/SnO_2_/CH_3_NH_3_PbI_3_/Spiro-OMeTAD/Au	22.0	1.084	0.642	15.29	
Li-doped	23.3	1.106	0.707	18.20	[87]
FTO/SnO_2_/(FAPbI_3_)_0.85_(MAPbBr_3_)_0.15_/Spiro-OMeTAD/Au	22.7	1.030	0.700	16.35	
Cl-doped	24.3	1.070	0.730	18.94	[90]
FTO/SnO_2_/Cs_0.1_FA_0.9_PbI_3_/Spiro-OMeTAD/Au	25.3	1.044	0.715	18.89	
NaCl-doped	25.5	1.069	0.758	20.68	[100]
FTO/SnO_2_/Cs_0.04_FA_0.74_MA_0.22_PbI_x_Br_y_Cl_3−x−y_/Spiro-OMeTAD/Au	23.4	1.132	0.750	19.89	
SbF_3_-doped	24.2	1.146	0.771	21.42	[103]
FTO/SnO_2_/(FAPbI_3_)_0.92_(MAPbBr_3_)_0.08_/Spiro-OMeTAD/Ag	22.13	1.034	0.515	11.83	
KCl-doped	23.02	1.100	0.761	18.54	[101]
ITO/SnO_2_/CsPbI_2_Br/Spiro-OMeTAD/MoO_3_/Ag	14.6	1.180	0.778	13.40	
KF-doped	14.8	1.310	0.792	15.39	[102]
ITO/SnO_2_/CH_3_NH_3_PbI_3_/Spiro-MeOTAD/Au	20.1	1.080	0.668	14.58	
Al_2_O_3_-doped	22.9	1.140	0.731	19.10	[106]
FTO/SnO_2_/CH_3_NH_3_PbI_3_/Spiro-MeOTAD/Au	23.1	1.089	0.691	17.35	
ZrF-doped	24.4	1.105	0.718	19.19	[104]
ITO/SnO_2_/PM6:Y6/MoO3/Ag	24.0	0.820	0.648	12.73	
PDINO-doped	26.4	0.825	0.687	14.97	[109]
ITO/SnO_2_/Cs_x_FA_y_MA_1-x-y_PbI_3-x_Cl_x_/Spiro-OMeTAD/Ag	21.8	1.070	0.740	18.74	
PEIE-doped	23.8	1.140	0.760	20.61	[110]
ITO/SnO_2_/Cs_0.04_FA_0.74_MA_0.22_PbI_x_Br_y_Cl_3−x−y_/Spiro-OMeTAD/Au	21.0	1.100	0.790	18.05	
PVP-doped	21.1	1.130	0.810	19.42	[111]
ITO/SnO_2_/Cs_0.05_FA_0.81_MA_0.14_PbI_2.55_Br_0.45_/Spiro-OMeTAD/Au	22.6	1.070	0.771	18.6	
PEG-doped	22.7	1.110	0.818	20.80	[108]
ITO/SnO_2_/Cs_0.05_FA_0.81_MA_0.14_PbI_2.55_Br_0.45_/Spiro-OMeTAD/MoO_3_/Au	23.1	1.070	0.778	19.15	
CQDs-doped	24.1	1.140	0.829	22.77	[62]
FTO/SnO_2_/(FAPbI_3_)_0.95_(MAPbBr_3_)_0.05_/Spiro-OMeTAD/Au	24.4	1.081	0.778	20.55	
N-doped	24.8	1.155	0.817	23.41	[92]
FTO/SnO_2_/(FAPbI_3_)_0.95_(MAPbBr_3_)_0.05_/Spiro-OMeTAD/Au	23.9	1.040	0.753	18.72	
F-doped	24.3	1.140	0.802	22.12	[93]
ITO/SnO_2_/(FAPbI_3_)_0.95_(MAPbBr_3_)_0.05_/Spiro-OMeTAD/Ag	21.8	1.150	0.732	18.41	
P-doped	22.6	1.140	0.767	19.72	[94]

**Table 2 materials-16-06170-t002:** Device structure and performance parameters of other inorganic ETLs.

ETL	Device Structure	J_sc_ (mA cm^−2^)	V_oc_ (V)	FF	PCE (%)	Ref.
TiO_2_	FTO/TiO_2_/MAPbI_3_/Spiro-OMeTAD/Au	23.31	1.172	0.765	20.90	[112]
ZnO	ITO/ZnO/PBDB-T:ITIC/MoO_3_/Ag	14.6	0.875	0.625	16.91	[113]
WO_x_	FTO/WO_x_/(FAPbI_3_)_1−x_(MAPbBr_3_)_x_/Spiro-OMeTAD/Ag	24.8	1.060	0.791	20.77	[114]
Nb_2_O_5_	ITO/NiO_x_/FA_0.85_MA_0.15_PbI_2.55_Br_0.45_/Nb_2_O_5_/Ag	22.7	1.083	0.745	18.28	[115]
SrTiO_3_	ITO/SrTiO_3_/Cs_0.07_FA_0.73_M_A0.20_PbI_2.53_B_r0.47_/Spiro-OMeTAD/Au	23.0	1.143	0.721	19.0	[116]
BaSnO_3_	FTO/BaSnO_3_/MAPbI_3_/Spiro-OMeTAD/Ag	15.7	0.984	0.628	9.65	[24]
CeO_x_	ITO/NiO_x_/MAPbI_3_/CeO_x_/Au	22.5	1.010	0.754	17.47	[117]
Cr_2_O_3_	FTO/Cr_2_O_3_/Cs_0.05_(MA_0.17_FA_0.83_)_0.95_Pb(I_0.83_Br_0.17_)_3_/Spiro-OMeTAD/Au	21.3	1.098	0.695	16.23	[18]
In_2_O_3_	FTO/In_2_O_3_/CH_3_NH_3_PbI_3_/Spiro-OMeTAD/Au	19.3	1.070	0.678	13.97	[118]
Fe_2_O_3_	FTO/Fe_2_O_3_/CH_3_NH_3_PbI_3_/Spiro OMeTAD/Au	16.6	0.650	0625	10.78	[119]
Al_2_O_3_	FTO/Al_2_O_3_/(FAPbI_3_)_1−x_(MAPbBr_3_)_x_/Spiro-OMeTAD/Ag	22.8	1.060	0.670	16.23	[120]
Eu_2_O_3_	ITO/NiO_x_/CsPbI_2_ Br/Eu_2_O_3_/PCBM/Bphen/Ag	15.28	1.140	0.776	13.47	[121]
Zn(In)O	ITO/NiO_x_/Cs_0.05_(FA_0.83_MA_0.17_Pb(I_0.83_Br_0.17_))_0.95_/Zn(In)O/Al	22.9	1.010	0.563	16.20	[122]
Ti(Nb)O_x_	ITO/NiO_x_/MAPbI_3_/PCBM/Ti(Nb)O_x_/Ag	21.9	1.070	0.790	18.49	[123]
Zn_2_SnO_4_	ITO/Zn_2_SnO_4_/PCBM/CH_3_NH_3_PbI_3_/Spiro-OMeTAD/Ag	21.2	1.070	0.617	14.50	[124]
N-doped SnO_2_	FTO/N-dopedSnO_2_/(FAPbI_3_)_0.95_(MAPbBr_3_)_0.05_/Spiro-OMeTAD/Au	24.8	1.155	0.817	23.41	[92]

## Data Availability

The authors do not have permission to share data.

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
