# Peer review of "Recent Advances of Doped SnO_2_ as Electron Transport Layer for High-Performance Perovskite Solar Cells"

_materials, 2023, doi:10.3390/ma16186170_

Round 1

Reviewer 1 Report

(1) "first time" may be overused in the article. In my knowledge, the following "first time" statements are not proper since those discussions/insights are available in the literature. Please revise these statements, or provide scientific evidence to support these "first time" if you do have concrete reasoning.

Line15: "we provide for the first time a comprehensive insight into the factors that specifically influence the ETL in PSC."

Line17: "the general operating principles, as well as the suitability level of doping in SnO2, are first elucidated..."

Line94: "the primary factors affecting the ETL properties are listed for the first time..."

Line95: "the detailed mechanisms for doping selection based on their suitability for different synthetic techniques are explored for the first time..."

(2) The title of Table 1 is "...their contribution to the improvement...". However, no comparison data are provided in the table to show the "improvement". I suggest the authors to add device performance data without dopants to show the "improvement" after adding the dopants.

Furthermore, device structure should be added to each data to allow comparison of device performance appeared in different reference papers.

Author Response

Thank reviewer 1 for the straightforward summary. We will address all the issues raised by reviewer 1 as much as possible. Please see the attachment.

Reviewer 2 Report

The review paper introduced the role of SnO2 in the perovskite solar cell development. Overall, the review is well-organized and includes very rich knowledge of doped undoped SnO2 contribution in PSC. Minor comments are as follows:

1- The PSC introductory part is quite extended. It should be reduced, given that the core of the review is related to SnO2 contribution in PSC.

2- More physical explanations for the existed defects according to the dopants with references support, are they substitutional defects or others?

3- A comparison between best doped-SnO2-based-PSC  and other oxides-PSC have to be mentioned clearly to support the enhancement of SnO2-PSC compared to other oxides.

4- Make sure of copyrights clearance of figures, especially SEM and TEM.

The review paper introduced the role of SnO2 in the perovskite solar cell development. Overall, the review is well-organized and includes very rich knowledge of doped undoped SnO2 contribution in PSC. It is worth to be published after few minor revisions, as sent to the authors.

Author Response

Thank reviewer 2 for the straightforward summary. Please see the attachment.

Reviewer 3 Report

This review paper focuses on the performance of SnO2 as the ETL of perovskite social cell. Relevant background regarding the principle of PSC and ETL were covered in sufficient detail. The factors that affect the SnO2 properties and device performance were discussed comprehensively. This work has merit to the PSC community and can be published without change. 

Author Response

Thanks reviewer 3 for the straightforward summary. 

Reviewer 4 Report

The authors provide an overview of the tin based materials used in solar cells, identifiying key problems and general research directions. Outlook and perspective reflects upon potential future direction of research.

Author Response

Thank reviewer 4 for the straightforward summary
